# Dendrochronological evidence for long-distance timber trading in the Roman Empire

**Mauro Bernabei**[1]*, **Jarno Bontadi**[1], **Rossella Rea**[2], **Ulf Büntgen**[3,4,5,6], **Willy Tegel**[7,8]

**1** CNR-IBE, Institute for BioEconomy, National Research Council, S. Michele all'Adige, TN, Italy,
**2** Soprintendenza Speciale Archeologia, Belle Arti e Paesaggio di Roma, Italy, **3** Department of Geography, University of Cambridge, Cambridge, United Kingdom, **4** Swiss Federal Research Institute (WSL), Switzerland, **5** Global Change Research Centre (CzechGlobe), Czech Republic, **6** Department of Geography, Faculty of Science, Masaryk University, Czech Republic, **7** Institute of Forest Sciences, Chair of Forest Growth, Albert-Ludwigs-University Freiburg, Freiburg, Germany, **8** Amt für Archäologie, Kanton Thurgau, Frauenfeld, Switzerland

* mauro.bernabei@ibe.cnr.it

**Data Availability Statement:** All relevant data are within the paper and Supporting Information files.

**Funding:** WT received funding from the German Research Foundation (DFG, TE 613/3-2). UB

## Abstract

An important question for our understanding of Roman history is how the Empire's economy was structured, and how long-distance trading within and between its provinces was organised and achieved. Moreover, it is still unclear whether large construction timbers, for use in Italy, came from the widespread temperate forests north of the Alps and were then transported to the sparsely-wooded Mediterranean region in the south. Here, we present dendrochronological results from the archaeological excavation of an expensively decorated portico in the centre of Rome. The oak trees (*Quercus* sp.), providing twenty-four well-preserved planks in waterlogged ground, had been felled between 40 and 60 CE in the Jura Mountains of north-eastern France. It is most likely that the wood was transported to the Eternal City on the Saône and Rhône rivers and then across the Mediterranean Sea. This rare dendrochronological evidence from the capital of the Roman Empire gives fresh impetus to the ongoing debate on the likelihood of transporting timber over long distances within and between Roman provinces. This study reconstructs the administrative and logistic efforts required to transport high-quality construction timber from central Europe to Rome. It also highlights an advanced network of trade, and emphasises the enormous value of oak wood in Roman times.

## Introduction

"*Mille praetera sunt usus earum, sine quis vita degi non possit*", ("Wood has thousands of uses, and without it, life would not be possible", Pliny the Elder: Naturalis Historia XVI, 1–5). With this declaration, Pliny (23/24-79 CE) points out the value wood had for the Romans. Wood was important for any aspect of everyday life, ranging from the construction of buildings [1] to heating systems [2], and from shipbuilding [3] to metalworking [4]. In Latin, the distinction between firewood, *lignum*, and construction timber, *materia*, is indicative in this respect. The current Spanish word for wood is *madera*. However, in other languages the word "material" (in English) or *Material* (in German) has taken on a more general meaning, signifying "matter" or "substance". Basically, wood was so important for the Romans that they considered it as "material" in the modern, English, sense of the word.

received funding from the Czech Republic Grant Agency (17-22102s).

**Competing interests:** The authors have declared that no competing interests exist.

In Rome, timber requirements were immense [5,3]. The demand for timber led to the rapid depletion of the woodlands surrounding the capital and in much of the Apennines. As the Empire expanded, timber cutting continued abroad: in Pliny's time (1st century CE), some of Algeria's forests rich in sandarac trees (*Tetraclinis articulata*), a wood particularly appreciated by the Romans, had already been fully exploited so that its timber supply shifted to Morocco [1]. And Emperor Hadrian created an imperial forest, by fencing off the cedar of Lebanon woodland and marking its perimeter with inscribed boundary stones, in order to conserve those woods [6].

A great variety of tree species was available in Rome in large quantities: ebony (*Diospyros* spp.), cedar (*Cedrus* spp.), box (*Buxus sempervirens* L.), terebinth (*Pistacia terebinthus* L.), holm oak (*Quercus ilex* L.) and many others. Patrician houses commonly contained a wide choice of wood [7] and were adorned with other precious material like gold or ivory. For the construction of buildings, silver fir (*Abies alba*) was the preferred tree species. Vitruvius himself, in his treatise on architecture (*De Architectura*, II, chap. 9–10; 30–15 BCE), indicates the characteristics that make silver fir particularly valuable: its light wood and a large, regular stem. Archaeological finds in Pompeii and Herculaneum confirm this, where silver fir was the most common construction timber [8], followed by oak wood, which is heavier than silver fir and has a less regular stem, especially in the case of trees from the Apennines [4]. However, oak is stronger, harder and much more durable than fir. These characteristics made oak less suitable for providing long roof beams or roof trusses but perfect for all kinds of foundations in contact with the ground. Despite our understanding of the many uses of wood in Roman times, detailed insight into long-distance timber trading, the preferred tree species used and its sources of supply is still limited [1,9].

At the same time, recent advances in dendrochronology have made important contributions to archaeological research [10]. Given the right conditions, wood can be dendrochronologically dated to the calendar year [11,12]. Moreover, tree-ring research can determine the wood's provenance [13], and sometimes it may even help to identify political and economic networks of commercial trade [14]. Unfortunately, in the Mediterranean region, the necessary conditions for dendrochronological analysis are rarely given [15,16]. Wood is preserved over a long period of time only in very humid or very dry locations, at very low temperatures, in contact with metal or in the form of charcoal [7,17]. In Mediterranean archaeological excavations, pottery and iron, for example, are easily found but wood is rare, and often it only occurs as minute fragments bonded to metal [18].

Due to the scarcity of datable wood, only a few and so far unpublished multi-millennial reference chronologies exist for Italy. This circumstance effectively hampers the dendrochronological assessment of Roman timbers. Hence, most studies on Roman timber constructions refer to archaeological sites outside of Italy concerning, for instance, the dating of ships [19,20,21,22], barrels [23] and the reconstruction of trade routes [24,25].

This study is, therefore, the rare exception of a successful dendrochronological investigation of archaeological timbers in the city of Rome, which has allowed us to: a) date these timbers by the method of dendrochronological cross-dating, b) determine the timbers' geographical origin (provenance), and c) compare the dendrochronological results with those derived from historical and archaeological sources (multi-proxy comparison).

## Material and methods

### Archaeological evidence

During the construction of Rome's Metro line C (underground railway line) in 2014–16 CE, an archaeological excavation was carried out, covering an area of approximately 1,440 m$^2$, in

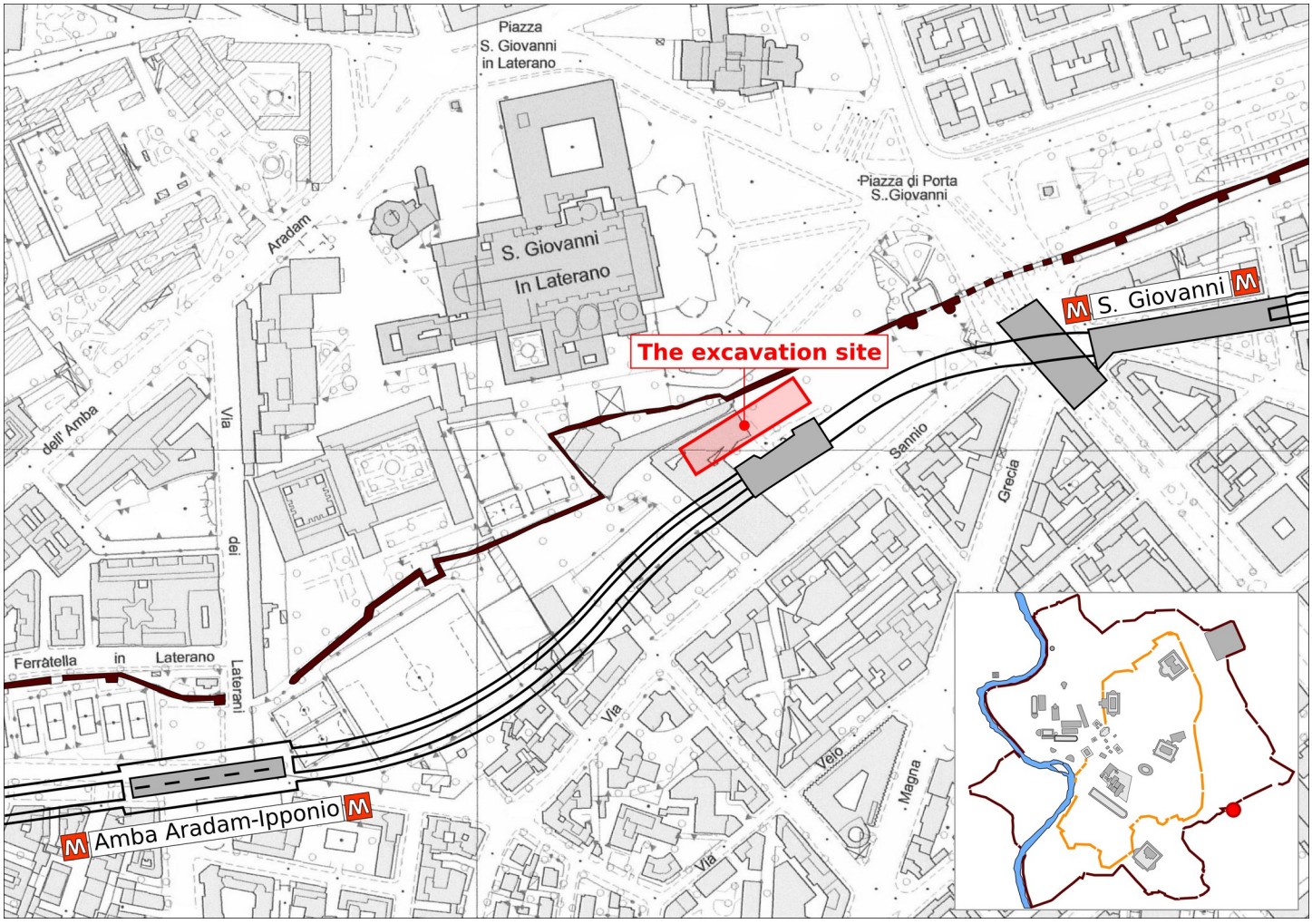

**Fig 1. The archaeological site.** The map shows the site of the excavation in Rome, between the ancient Aurelian walls (marked by a black line) and Rome's Metro line C (underground railway line), near the Basilica of San Giovanni in Laterano. Map sourced from http://dati.lazio.it/catalog/it/dataset/carta-tecnica-regionale-2002-2003-5k-roma. On a map of ancient Rome at the bottom right corner, the excavation site is indicated by a red dot.

the gardens of via Sannio, next to the line. On this site, a total of twenty-four oak planks (*Quercus* sp.) were found under via Sannio (Fig 1), near the Basilica of San Giovanni in Laterano, just outside the ancient Aurelian walls. These planks had been part of the foundations of a richly-decorated portico (Fig 2), belonging to a vast and wealthy property [26]. All samples were well-preserved, as they were saturated with water. In many cases, the vessels of the wooden planks were filled with hard, glassy, translucent whitish mineral deposits deriving from the site. Six planks come from the south-eastern foundations (stratigraphic unit, s.u., 1141, Table 1), and consist of two non-continuous rows of horizontal planks, shored up by round posts (Figs 2 and 3). Most of the planks were about 3.60 m long, with the shortest one having a length of 1.15 m. The round posts were 60–66 cm long, with a diameter of 3.5–8 cm. Sixteen planks are from the north-western foundations (s.u. 1143, Fig 3), making up four rows of horizontal planks (1.67–3.57 m, though mostly about 3.50 m long; 25–30 cm wide). These planks were also shored up with 65 cm long posts of 4–9 cm diameter. Another sample, code A (Table 1) (s.u. 1286), belonged to the foundations of an older construction on the same site. Finally, sample B (s.u. 1383) was a support for a small bridge over a watercourse.

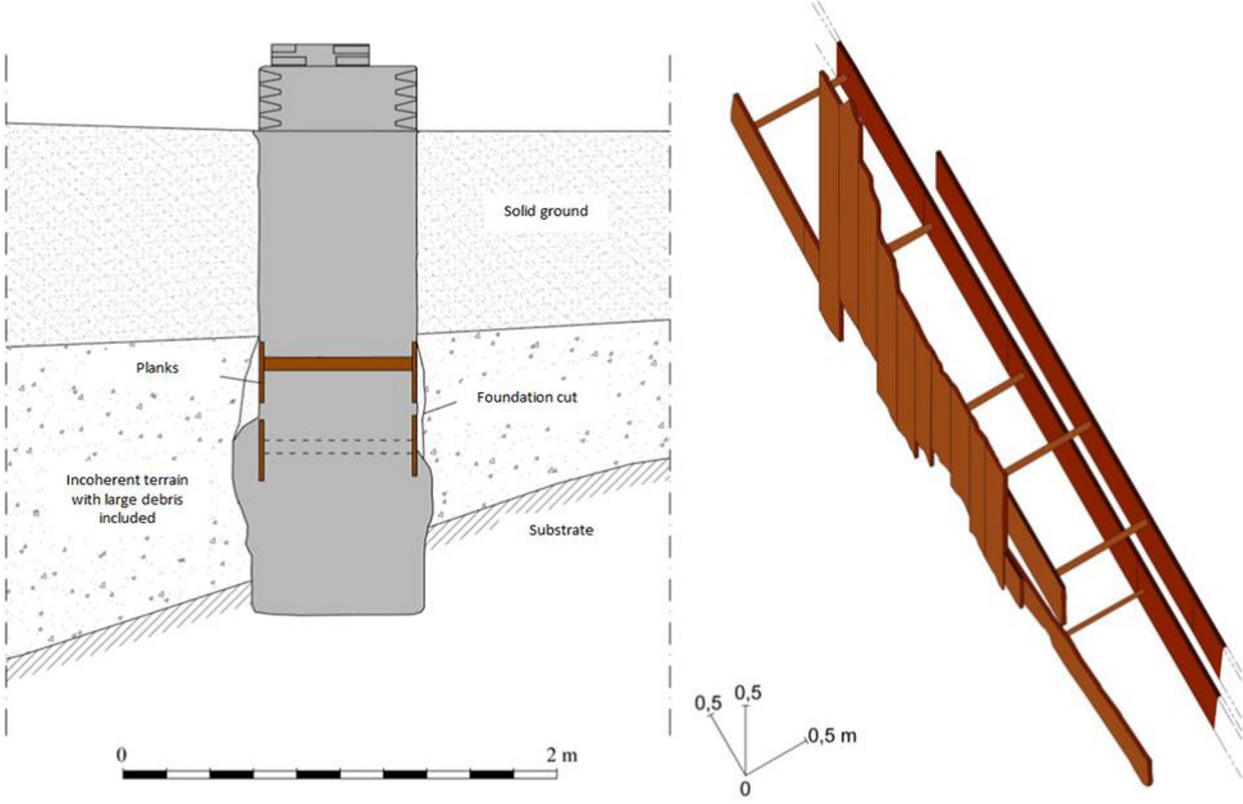

**Fig 2. The foundation of the portico.** Schematic section of the foundations of the south-eastern lateral portico (u.s 1141–1178) [26].

A description of the archaeological site is found at the San Giovanni Station Museum in Rome, an exhibition showing the stratigraphy of the excavation and some of the objects found. All necessary permits for the described study were obtained from the Soprintendenza Speciale Archeologia, Belle Arti e Paesaggio di Roma.

## Dendrochronological dating

Cross-sections of each plank were carefully prepared with a scalpel, and chalk was rubbed into the clean surface in order to highlight the anatomical tree-ring pattern. Tree-ring widths (TRW) were measured perpendicularly to the ring tangent, using a LINTAB device (LINear TABle, RinnTech, Heidelberg, Germany), with an accuracy of 0.01 mm. The TSAP-Win program [27] was used to record measurements and create individual TRW series. Where more than one TRW series was obtained from the same sample, these series were averaged. In oak trees, the ring border is often deformed by large parenchymatous rays, hence, the measurement direction required continuous adjustment. Where a cross-section also contained the pith, both radii were measured to exclude possible deformed rings. Another aspect that will affect ring-measurement precision on planks, though not on stem cross-sections, are anomalous, eccentric growth rings that are typical of oak. Each individual TRW series was compared visually and statistically (using PAST4 [28]) with all other individual series to ensure measurement accuracy and to identify possible xylological anomalies such as false or missing rings, or deformation caused by large parenchymatous rays. Finally, a mean TRW chronology was obtained by averaging TRW series with significant cross-correlation values ($T_{BP} > 4$, see details below).

**Table 1. Samples from the archaeological excavation.**

| Sample code | Stratigraphic unit | Position |
|---|---|---|
| A | M1286 | Oldest foundation |
| B | 1383 | Canal bridge |
| C1 | 1359 (1143) | NW Foundation |
| C2 | 1359 (1143) | NW Foundation |
| C3 | 1359 (1143) | NW Foundation |
| C7 | 1359 (1143) | NW Foundation |
| C14 | 1359 (1143) | NW Foundation |
| C17 | 1359 (1143) | NW Foundation |
| C20 | 1359 (1143) | NW Foundation |
| C22 | 1359 (1143) | NW Foundation |
| C24 | 1359 (1143) | NW Foundation |
| C25 | 1359 (1143) | NW Foundation |
| C27 | 1359 (1143) | NW Foundation |
| C29 | 1359 (1143) | NW Foundation |
| C31 | 1359 (1143) | NW Foundation |
| C32 | 1359 (1143) | NW Foundation |
| C37 | 1359 (1143) | NW Foundation |
| C43 | 1359 (1143) | NW Foundation |
| C53 | 1178 (1141) | SE Foundation |
| C54 | 1178 (1141) | SE Foundation |
| C56 | 1178 (1141) | SE Foundation |
| C58 | 1178 (1141) | SE Foundation |
| C59 | 1178 (1141) | SE Foundation |
| C60 | 1178 (1141) | SE Foundation |

Each comparison was based on two statistical parameters: a) $T_{BP}$ and $T_{HO}$: t-value adapted to time-series by Baillie and Pilcher [29] and Hollstein [30]; b) Glk: *Gleichläufigkeit* and its value *Gleichläufigkeitswert*, as discussed by Eckstein and Bauch [31]. Glk is a non-parametric test, which, by comparing two time series at a given time-interval, represents the percentage of agreement between the growth sign (+ or -) from one year to the next. The significance level of the correlation coefficient is assessed at p = 0.05, 0.01 and 0.001, here indicated as *, ** and ***; c) overlap: the number of rings compared, to which the statistical tests refer.

## Results

All samples belong to *Quercus* sp, subgenus *Quercus*. The wood's decay rendered any microscopic distinction between the sections *Quercus* and *Cerris* [32] impossible.

Visual examination of the planks did not reveal any traces of previous usage, such as old dowels, engravings, marks from previous work or traces of insects. Only axe marks from very accurate cutting and wood preparation (Fig 3) were found, indicating that the timbers used were cut specifically for this construction.

A total of thirteen planks was successfully dated. The individual TRW series ranged from 27 to 284 years, with a mean segment length of 135 rings and an average growth increment of 1.1 mm. Four of twenty samples had more than 250 rings (Table 2). The TRW patterns of thirteen series were very similar, which is reflected in their significant, positive inter-series correlation (Rbar = 0.4, Table 2 and Fig 4). These individual TRW series form the mean site chronology RMC1 (Roman Metro Chronology 1) that spans 320 years (Fig 4). Given the lack

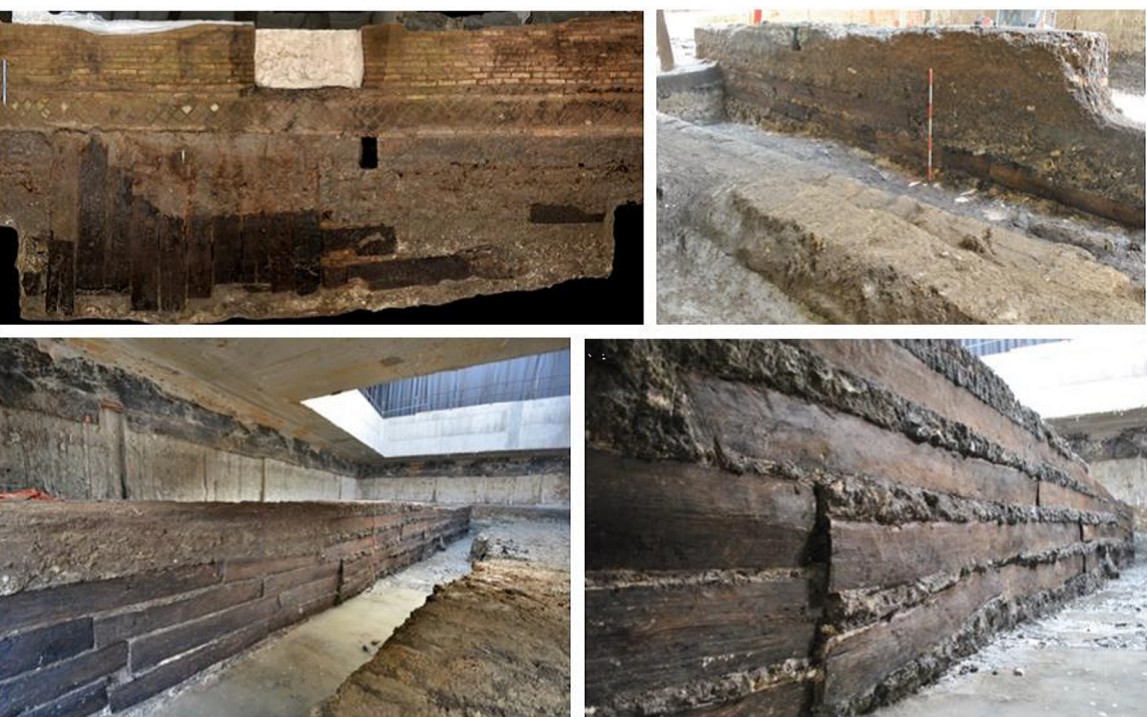

**Fig 3. Oak planks *in situ*.** At the bottom right, axe marks are visible on the planks [26].

of a solid Italian oak reference chronology for the Roman period, RMC1 was compared with several floating Italian chronologies in the Roman period (Martinelli N., unpublished data). However, no significant correlation was found. Hence, RMC1 was further compared with central European oak chronologies [30,33], which indicates a statistically significant correlation for the years 279 BCE-40 CE.

Having successfully dated the Roman oak planks against central European reference chronologies (above), a closer look at regional reference chronologies points to the timber's provenance in eastern France, i.e. Alsace (Upper Rhine Valley), Lorraine, Champagne and Burgundy.

Indeed, high correlation values were obtained with a reference chronology for the Alsace region ($T_{HO}$ 6.0), followed by Burgundy ($T_{HO}$ 5.6). By contrast, regions located further north such as Lorraine and Champagne show less significant correlations (Table 3). RMC1 has the highest correlation values ($T_{HO}$ and $T_{BP}$ 6.1) with the Moissey "La Tuilerie" Chronology, Dép. Jura [34], which confirms that the timber originates from the Jura mountain range (Fig 5).

Sapwood was identified in eight samples (Table 2). The presence of sapwood was also microscopically verified by the absence of tyloses in large vessels of the porous rings. Even if a sample only contained sapwood remains, this narrows down the estimate of the tree's felling date. As a general rule, mature oaks develop between 10 and 30 sapwood rings [30]. Hence, the felling date can be estimated with a precision of ±10 years. On the basis of these calculations, and considering that sapwood estimates may vary geographically, all dated trees were felled between 40 and 60 CE (Table 2).

## Discussion

The Romans based their hegemony on an imposing road system, which enabled long-distance trade and the massive exploitation of resources in the regions they controlled [37]). Metal,

**Table 2. Dendrochronological correlation values of each individual TRW series against the mean chronology of all other samples.**

| Code | Stratigraphic unit | Distance to pith (cm) | Sapwood[a] (no. of rings) | Tree rings (no.) | Dating (last ring) | $T_{BP}$; Glk (%)*** |
|---|---|---|---|---|---|---|
| A | M1286 | 10 | 35 | 81 | //[b] | |
| B | 1383 | 5 | 0 | 44 | // | |
| C1 | 1359 (1143) | 3 | 15 | 284 | **40 CE** | 12.20; 65.50*** |
| C2 | 1359 (1143) | >10 | 0 | 278 | // | |
| C3 | 1359 (1143) | 5 | 0 | 283 | **3 CE** | 9.21; 65.90*** |
| C7 | 1359 (1143) | 10 | 0 | 269 | **14 CE** | 7.30; 61.90*** |
| C14 | 1359 (1143) | 10 | 0 | // | // | |
| C17 | 1359 (1143) | >10 | 12 | 174 | **40 CE** | 7.36; 63.90*** |
| C20 | 1359 (1143) | 5 | 0 | // | // | |
| C22 | 1359 (1143) | 1 | 1(?) | 100 | **23 CE** | 5.77; 75.00*** |
| C24 | 1359 (1143) | 10 | 0 | 60 | **54 BCE** | 4.28; 62.50* |
| C25 | 1359 (1143) | 10 | 0 | 78 | // | |
| C27 | 1359 (1143) | >10 | 0 | 102 | **84 BCE** | 5.36; 69.10*** |
| C29 | 1359 (1143) | 0 | 1(?) | 114 | **21 CE** | 4.72; 68.40*** |
| C31 | 1359 (1143) | 5 | 0 | // | // | |
| C32 | 1359 (1143) | 1 | 0 | 69 | **15 CE** | 4.52; 62.30* |
| C37 | 1359 (1143) | 8 | 10 | 97 | **34 CE** | 8.52; 64.40** |
| C43 | 1359 (1143) | >10 | 0 | 27 | // | |
| C53 | 1178 (1141) | 1 | 0 | 61 | **11 BCE** | 4.76; 73.00*** |
| C54 | 1178 (1141) | >10 | 20 | 81 | **30 CE** | 4.08; 51.90* |
| C56 | 1178 (1141) | 3 | 1(?) | // | // | |
| C58 | 1178 (1141) | >10 | 0 | 141 | // | |
| C59 | 1178 (1141) | 2 | 0 | 59 | **4 BCE** | 5.15; 71.20*** |
| C60 | 1178 (1141) | 0 | 0 | 28 | // | |

[a]Sapwood is the outer layer of recently formed wood, usually of lighter colour, between the heartwood and the bark, containing the functioning vascular tissue.

[b]// indicates: no results

pottery, marble and many other luxury goods were transported, regardless of distance and possible geographical barriers. This is fairly easy to imagine in the case of luxury goods and non-perishable food but it is more complicated to achieve for imposing blocks of marble for Roman squares or with the wild animals to be killed by gladiators in Roman amphitheatres [38]. Hence, the infrastructures of the Roman Empire, combined with advanced logistical skills, require our admiration and respect to the present day.

The principal routes for commercial transport in Roman times are well known [37,39], and there are detailed descriptions of the Romans' means of acquisition as well as the exact origin of the goods [1,38]. However, very little is known about timber trading to Rome and the existence of commercial routes for this purpose [40,9]. Particularly valuable tree species, or those used as status symbols, like the sandarac tree (*Tetraclinis articulata*) or ebony (*Diospyros* sp.), were imported but there are no documents in support of this, and there is no evidence or any proof at all that would indicate long-distance timber trading to Rome for construction purposes [9]. We do know that wood was transported over long distances after the fall of the Roman Empire. Timber, coming from the Alps and already in the form of long beams, was shipped right across the Mediterranean Sea to be used in Palestine [14].

Our research has shown that in Roman times central European wood was used for construction purposes in central Rome, and a commercial route of transport has been identified

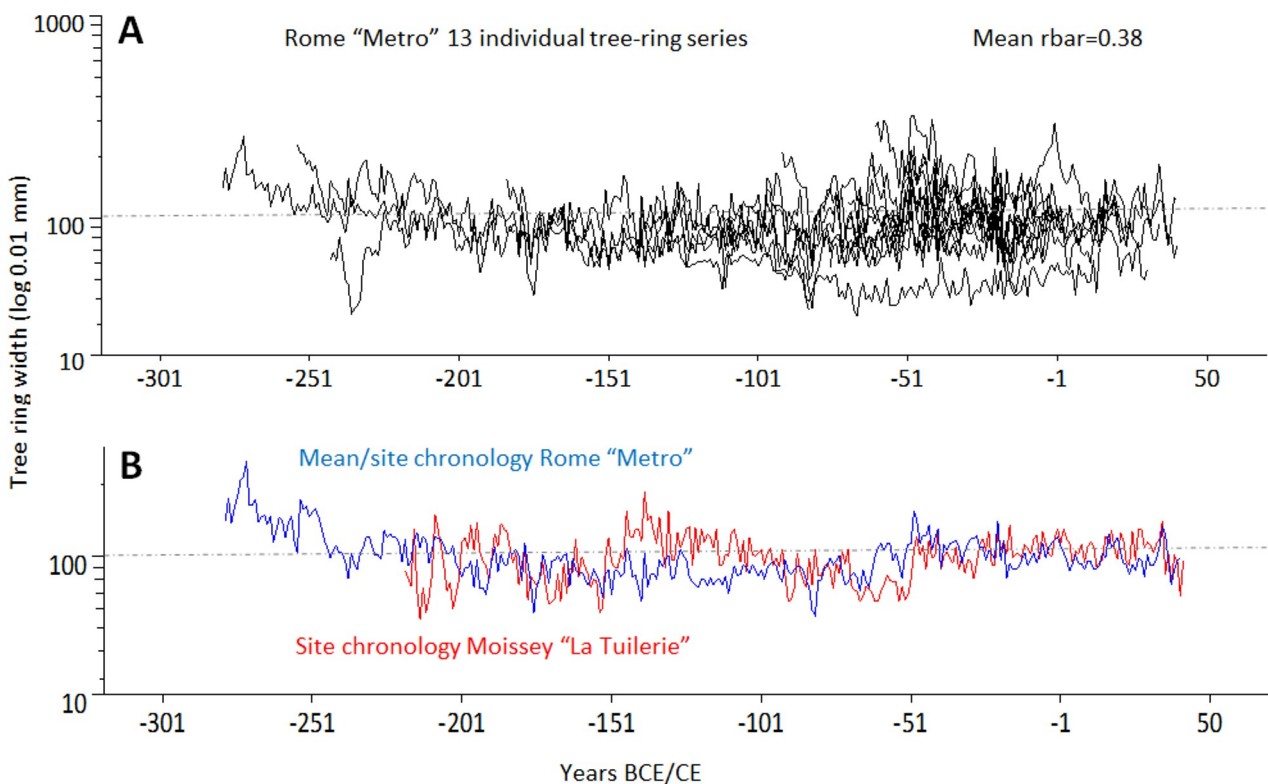

**Fig 4. The TRW visual comparison.** Visual comparison of all dated individual Roman oak TRW series (A), and of the mean "Rome Metro Site Chronology" (RMC1, in red) with the "Moissey-La Tuilerie Site Chronology" (B) from the French Jura, demonstrate the highest correlation values for the year 40 CE ($T_{BP}$ = 6,57, Glk 64,40***, see Table 3).

for this scope. In fact, a comparison between the Roman oak planks examined and existing site chronologies (Table 3) does not leave any doubt that the provenance of the oaks used for building the patrician porch is today's north-eastern French region of the Jura. This statement is based on the following results:

1. There is no statistically significant correlation between the Rome Metro Chronology (RMC1) and any dendrochronological series of Italian oak wood from Roman times. Although it is difficult to find wooden samples in archaeological excavations in Italy, numerous valid dendrochronological series have, in fact, been obtained for ancient times (Martinelli N., Wazny T. pers. com.) but they had to be radiocarbon-dated.

2. Comparing the Roman oak chronology with non-Italian site chronologies has, on the other hand, produced increasingly significant results as one moves towards Central Europe, where the highest correlation values were obtained for site chronologies from north-eastern

**Table 3. Correlation values of the Rome Metro Site Chronology (RMC1) in comparison with European reference chronologies for the year 40 CE.**

| Region/Site | Reference | Overlap | $T_{BP}$ | $T_{HO}$ | Glk (%) |
|---|---|---|---|---|---|
| Moissey-La Tuilerie (Jura) | Charlier 2001 [34] | 260 | 6.1 | 6.1 | 64.8*** |
| Alsace | Tegel *et al.* 2016 [35] | 260 | 5.5 | 6.0 | 65.6*** |
| Lorraine | Tegel *et al.* 2016 [35] | 260 | 3.3 | 4.1 | 55.6* |
| Champagne | Tegel *et al.* 2016 [35] | 260 | 3.6 | 4.2 | 61.0** |
| Burgundy | Lambert and Lavier 1991 [36] | 260 | 4.7 | 5.6 | 58.5** |

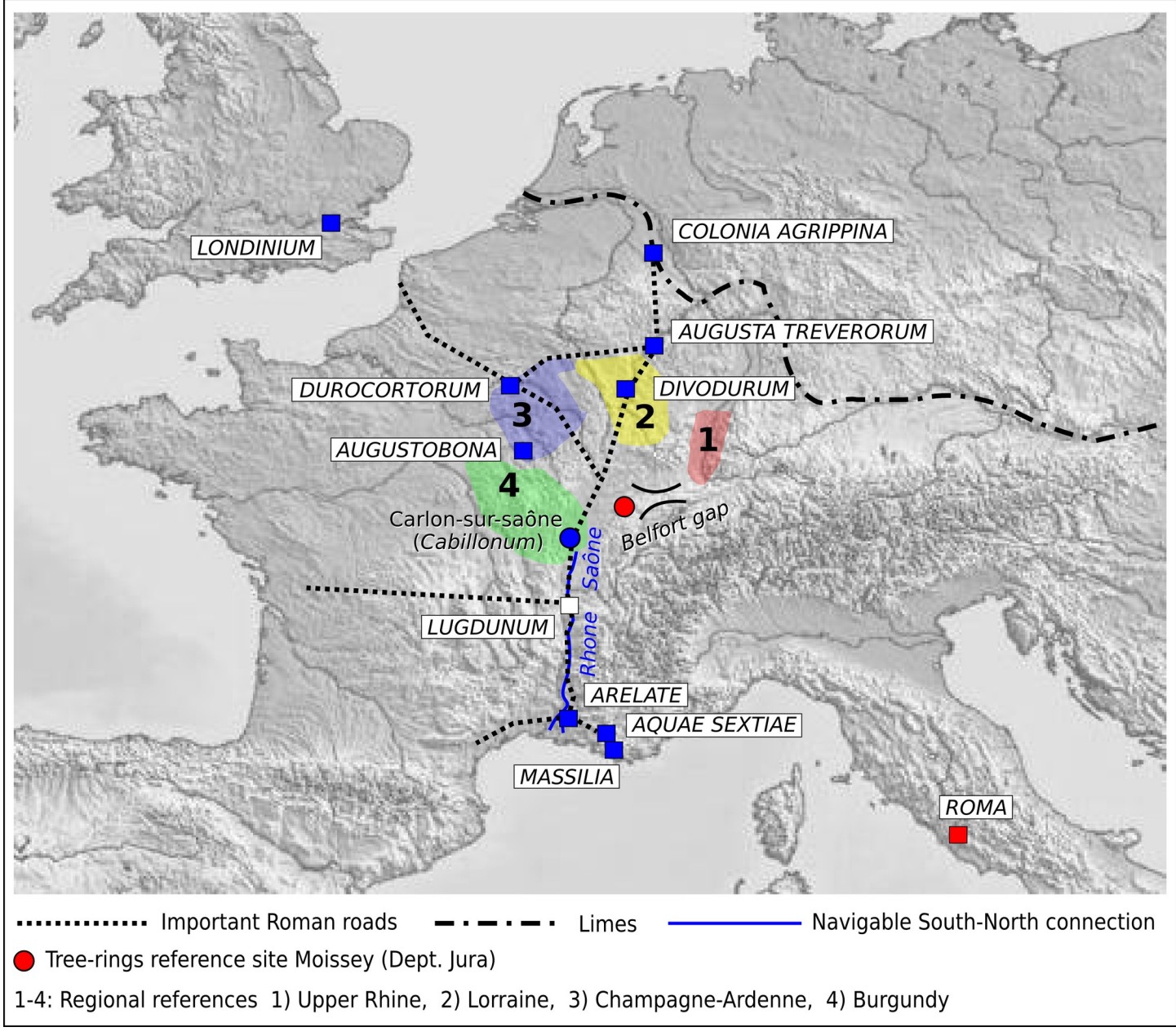

**Fig 5. The timber road.** Map of Roman provinces in today's France and Germany, with the probable provenance of the Roman oak Metro samples. Some important Roman towns are indicated (*Colonia Agrippina* = Cologne; *Augusta Treverorum* = Trier; *Divodurum* = Metz; *Dorocortorum* = Reims; *Augustobona* = Troyes; *Lugdunum* = Lyon; *Arelate* = Arles; *Aquae Sextiae* = Aix-en-Provence and *Massilia* = Marseilles), as well as the regions where the reference chronologies (Table 3) come from and also the rivers (Saône and Rhône) leading to the Mediterranean Sea. Map sourced from https://mapswire.com/europe/.

France. These correlation values are even better if one considers that the Rome Metro Chronology only consists of thirteen individual time series, whose total length of 320 tree rings is reduced to 280 rings if only the well-replicated part of the chronology is considered (Table 3, Fig 4).

It is well-known that Central European oak growing in an oceanic or continental climate is very different from that in the Mediterranean [41]. Hence, Mediterranean site chronologies

from Roman times cannot be successfully cross-dated with the long central European oak reference chronologies [41].

As our TRW series cross-dated best with the Moissey site chronology in the French Jura, it can be assumed that the oaks originate from this region, which is also close to the most important–and in great part navigable–commercial trade routes of ancient Gaul [40]. Moissey is located 60 km north-east of Chalon-sur-Saône (*Cabillonum*), at the onset of the Belfort Gap (Fig 5), which separates the drainage basins of the rivers Rhine and Rhône (Doubs, Ognon and Saône) between the low mountain ranges of Vosges and Jura. Due to its outstanding topographic location, this was an important area, connecting the provinces of Gaul and *Germania Superior*. The timber was most likely transported by road or rafted and floated downstream on the Saône river up to Chalon-sur-Saône, an important commercial centre of Gaul and, at the time, the administrative headquarters of a river fleet [40]. From this point, the Saône is then navigable southwards, down to the river Rhône. At the confluence, there is Lyon (ancient *Lugdunum*), the most important commercial and administrative centre of the province of Gaul (*Gallia Lugdunensis*). Even large ships could transport the timber to Rome's ancient port of Ostia, via the river Rhône and the Ligurian Sea, and then up the river Tiber [42].

The dendrochronological dates are confirmed by archaeological finds such as large quantities of ancient, red Roman pottery, which contains many marked fragments that were found underneath the pavement and date to the same time period.

It is interesting to note how the wooden samples that contain a few sapwood rings are correctly aligned towards the end of the site chronology. This proves [19] that the samples belong to a single wood lot and that, therefore, they have the same provenance.

The samples' high reciprocal correlation values further confirm this, which has also led to the hypothesis that some planks may come from the same tree. Samples C1 and C3, in particular, have $T_{BP}$-values above 8 (8.15), if some external rings are excluded from the calculation. Similar correlation values are also found for samples C7 and C37 and the $T_{BP}$-values between individual tree-ring series and the site chronology of the remaining samples are rather high (Table 2).

Although four oak samples have more than 250 rings (Table 2), no sample extended from pith to bark, indicating that some of the oaks must have been fairly long-lived. Other trees from the same lot have considerably fewer rings. This might indicate that the trees came from near-natural oak woodlands, with an uneven age structure and very little human disturbance.

The vast forests in the northern provinces would have supplied seemingly endless raw material that, during the initial occupation of the new territories, was locally used to construct encampments and to supply the necessary war machinery, but at the end of the war was also employed for the upkeep of various strategic fortifications of the Empire, the fleet and for construction purposes [9]. Cutting the trees, stockpiling and working, and then transporting the timber; everything had to be organised in such a way that the half-prepared timber travelled safely from the French woodlands to the heart of Rome.

The felled trees were most likely split to make planks on site and were further processed by axes or adzes, as proven by characteristic toolmarks (Fig 3). Their plain surface indicates the use of freshly cut oaks. Also for practical reasons, we assume that the planks were prepared before the timber was transported over long distances. The dimension and weight of the oak planks suggest river and maritime rather than land transport. A dense network of waterway trade routes were intensively used to connect the Roman Empire.

Considering the distances, calculated to be over 1700 km, the timber's dimensions, road transport with all the possible obstacles along the way, floating the timber down rivers and finally shipping it across the sea, the logistic organisation of the Romans must have been formidable. Even more so as, in this case, the buried oak planks would not even have been visible, being part of the foundations.

It would probably have been more difficult, and less convenient, to find solid oak with the necessary characteristics in Apennine forests, rather than making them arrive from the occupied provinces, where a large workforce and the abundance of raw material made it easier to obtain wood of the best quality. In the province of Gaul, it was only necessary to manage the workforce but that was organised by Roman logistics.

## Conclusions

Although enormous quantities of wood were used in Roman times, south of the Alps dendrochronological analysis is difficult because few remains have survived intact [15]. So far, dendrochronological dating of southern Alpine oak timbers has proved to be difficult due to the lack of long reference chronologies. Here, the first successful cross-dating results of thirteen, on average 3.6 m long oak planks, from the foundations of a patrician villa in Rome, are presented. The planks are from up to 300-year old oak trees, felled between 40 and 60 CE in north-eastern France, possibly in near-natural woodlands of the Jura Mountains. The timber was rafted and floated down the principal waterways of the province of Gaul, the Saône and Rhône rivers, and then crossed the Mediterranean Sea to be finally taken up the river Tiber to Rome. For the first time, the use of oak trees grown in the Roman provinces north of the Alps has been proved for the construction of buildings in ancient Rome. The transport routes have been reconstructed on the basis of still existing waterways.

## Supporting information

**S1 Table. Thirteen individual tree ring series from Rome and the resulting mean chronology.** Species: *Quercus* sp. Tucson format.
(PDF)

## Acknowledgments

The authors thank Christophe Perrault and Nicoletta Martinelli for providing reference chronologies; Katarina Čufar, Peter Ian Kuniholm, Lars-Ake Larsson, Petra Ossowski Larsson, Charlotte Pearson and Tomasz Wazny for their suggestions and results; Nicoletta Saviane for documents and information regarding the archaeological excavation. MB thanks Alessandro Bezzi from Arc-Team for his fundamental help and suggestions.

## Author Contributions

**Conceptualization:** Mauro Bernabei, Rossella Rea, Ulf Büntgen.

**Data curation:** Mauro Bernabei, Jarno Bontadi.

**Investigation:** Mauro Bernabei, Jarno Bontadi, Willy Tegel.

**Project administration:** Rossella Rea.

**Resources:** Willy Tegel.

**Supervision:** Mauro Bernabei, Ulf Büntgen, Willy Tegel.

**Validation:** Willy Tegel.

**Writing – original draft:** Mauro Bernabei, Jarno Bontadi, Rossella Rea.

**Writing – review & editing:** Mauro Bernabei, Ulf Büntgen, Willy Tegel.

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
