## [Decision Letter · Decision Letter 0]

12 Aug 2019

PONE-D-19-20970

DENDROCHRONOLOGICAL EVIDENCE FOR LONG-DISTANCE TIMBER TRADING IN THE ROMAN EMPIRE

PLOS ONE

Dear Dr. Bernabei,

Thank you for submitting your manuscript to PLOS ONE. After careful consideration, we feel that it has merit but does not fully meet PLOS ONE’s publication criteria as it currently stands. Therefore, we invite you to submit a revised version of the manuscript that addresses the points raised during the review process.

All three reviewers think that the manuscript is interesting for publishing in PLOS ONE. However, they require minor revision before it can be accepted for publication.

In addition, please note that manuscripts reporting paleontology and archaeology research must adhere to the PLOS ONE policies described at http://journals.plos.org/plosone/s/submission-guidelines#loc-paleontology-and-archaeology-research. Specifically, appropriate specimen numbers should be provided, and the specimens should be publicly deposited or available for replication of the study.

We would appreciate receiving your revised manuscript by Sep 26 2019 11:59PM. To enhance the reproducibility of your results, we recommend that if applicable you deposit your laboratory protocols in protocols.io, where a protocol can be assigned its own identifier (DOI) such that it can be cited independently in the future. For instructions see: http://journals.plos.org/plosone/s/submission-guidelines#loc-laboratory-protocols

We look forward to receiving your revised manuscript.

Kind regards,

Michal Bosela, Ph.D.

Academic Editor

PLOS ONE

Journal Requirements:

WT received funding from the German Research Foundation (DFG, TE 613/3-2). UB received funding from the Czech Republic Grant Agency (17-22102s).

3. In your manuscript, please provide additional information regarding the specimens used in your study. Ensure that you have reported specimen numbers and complete repository information, including museum name and geographic location.

For more information on PLOS ONE's requirements for paleontology and archaeology research, see "" ext-link-type="uri" xlink:type="simple">https://journals.plos.org/plosone/s/submission-guidelines#loc-paleontology-and-archaeology-research.""

You may seek permission from the original copyright holder of Figure 1 to publish the content specifically under the CC BY 4.0 license. 

If you are unable to obtain permission from the original copyright holder to publish these figures under the CC BY 4.0 license or if the copyright holder’s requirements are incompatible with the CC BY 4.0 license, please either i) remove the figure or ii) supply a replacement figure that complies with the CC BY 4.0 license. Please check copyright information on all replacement figures and update the figure caption with source information. If applicable, please specify in the figure caption text when a figure is similar but not identical to the original image and is therefore for illustrative purposes only.

Additional Editor Comments:

Please note that manuscripts reporting paleontology and archaeology research must adhere to the PLOS ONE policies described at http://journals.plos.org/plosone/s/submission-guidelines#loc-paleontology-and-archaeology-research. Specifically, appropriate specimen numbers should be provided, and the specimens should be publicly deposited or available for replication of the study.

Reviewers' comments:

Reviewer's Responses to Questions

**Comments to the Author**

1. Is the manuscript technically sound, and do the data support the conclusions?

Reviewer #1: Yes

Reviewer #2: Partly

Reviewer #3: Yes

2. Has the statistical analysis been performed appropriately and rigorously? 

Reviewer #1: Yes

Reviewer #2: Yes

Reviewer #3: Yes

3. Have the authors made all data underlying the findings in their manuscript fully available?

Reviewer #1: Yes

Reviewer #2: Yes

Reviewer #3: Yes

4. Is the manuscript presented in an intelligible fashion and written in standard English?

Reviewer #1: Yes

Reviewer #2: Yes

Reviewer #3: Yes

5. Review Comments to the Author

Reviewer #1: The study presents an original research on dendrochronology of oak from an ancient Roman construction recently excavated in Rome. To my knowledge this is the first successful dendrochronological dating of oak from the given period in Rome. In addition to dating of the construction, the investigation enabled to define the origin of wood and revealed information on wood transport in Roman time, which has not been reported in archived documents. The study presents an excellent collaboration between wood science (dendrochronology) and humanities (archaeology) and provides enrichment for both disciplines.

Specific comments:

Lines 54, 55: please give also scientific names of wood species ebony, cedar, box, terebinth, holm oak

Lines 81-86

You can also mention recent successful application of dendrochronology related to Roman period south of the Alps like

Cufar et al. 2014. Journal of Archaeological Science (Roman barge in the Ljublanica river…)

Cufar et al. 2019. Les / Wood (Research potential of wood of barrels from Roman water wells)

Figure 1. map on the Left and on the Right

On both maps please use larger font to mark an important location (e.g. San Giovanni in Laterano) in the current version of the maps the font is too small to read the inscriptions

Line147 …. Finally, a mean TRW sample chronology

Change to …. Finally, a mean TRW chronology…

Line 149 Each comparison is based on …

Change to …was based on…

Figure 5 – Can you add a scale bar (in kilometres)? Or somewhere in the text mention approximate distance between the area of timber source and Rome

Reviewer #2: I found this an interesting, informative paper that set the results in good context, allowing the reader to understand the importance of the findings to wider fields of study. It is well written. I have a minor problem with the specificity of the conclusion that trees were felled between 40 and 53 CE - which, unless I am missing something, is not fully supported by the evidence in Table 2. If one takes the quoted sapwood range of 10-30, then C1 could go to 55CE, C17 to 58CE and C37 to 54CE, and given that sapwood estimates vary a little geographically, I think it unwise to stick rigidly to 53CE as the end point - would it not be safer to say 60CE, but then comment about any justification you may have for trimming that outer point?

line 89 - the ref to Corona 1974 does not fit comfortably in the sentence as it stands, needs a little editing so that it is clear just what is meant - presumably Corona - in a rather old reference now, states how difficult results are to obtain in Rome? Suggest this is re-worded.

line 160 Cerris? are you talking Q cerris? Unclear.

Fig 4 - presumably all DATED individual Roman oak series?

Throughout - use decimal point in t values, 6.57 not 6,57 (inconsistent at present)

The following references are used in the text but do not appear in the References list:

Janssen et al 2017

Eackstein and Bauch 1969

Momigliano and Schiavona 1988

Rea 2001

Reviewer #3: Highly interesting paper - the first scientific evidence of Roman timber trade from north-eastern France to Rome and solid confirmation of Roman long-distance timber trade. This is a confirmation of high level of wood technology and wood-working skills of the Romans, and their enormous organizational level. A few of my suggestions of minor improvements concern the introductory part:

- page 3, line 71: reference to Buengten et al. 2018 is not the happiest in this context. There were numerous scholars before, who stressed annual precision of dendrochronological dating - Douglass, Huber, Becker, Eckstein, etc.;

- page 4, line 84: Guibal focused his research on shipwrecks found south of the Alps;

- page 4, lines 88-89: Corona's investigation published in 1974 was not very successful, and had nothing to do with this study.

In general I would like to suggest to be more careful with references, and to reduce their number. Not every sentence must be supported by references.

6. PLOS authors have the option to publish the peer review history of their article (what does this mean?). If published, this will include your full peer review and any attached files.

Reviewer #1: No

Reviewer #2: No

Reviewer #3: No

---

## [Author Response · Author response to Decision Letter 0]

17 Sep 2019

RESPONSE TO REVIEWERS

Reviewer #1 (R1): The study presents an original research on dendrochronology of oak from an ancient Roman construction recently excavated in Rome. To my knowledge this is the first successful dendrochronological dating of oak from the given period in Rome. In addition to dating of the construction, the investigation enabled to define the origin of wood and revealed information on wood transport in Roman time, which has not been reported in archived documents. The study presents an excellent collaboration between wood science (dendrochronology) and humanities (archaeology) and provides enrichment for both disciplines.

Author response (AR): we thank the Reviewer 1 for his nice words and the following improving observations.

Specific comments:

R1: Lines 54, 55: please give also scientific names of wood species ebony, cedar, box, terebinth, holm oak

AR: the suggestion has been accepted and the scientific name inserted in the revised text.

R1: Lines 81-86: You can also mention recent successful application of dendrochronology related to Roman period south of the Alps like

Cufar et al. 2014. Journal of Archaeological Science (Roman barge in the Ljublanica river…)

Cufar et al. 2019. Les / Wood (Research potential of wood of barrels from Roman water wells)

AR: the suggestion has been accepted and the references inserted.

R1: Figure 1. map on the Left and on the Right

On both maps please use larger font to mark an important location (e.g. San Giovanni in Laterano) in the current version of the maps the font is too small to read the inscriptions

AR: the suggestion has been accepted and a new Fig 1, clearer and better defined, has been changed. The new map is under CC-BY 4.0 license. Please visit: http://dati.lazio.it/catalog/it/dataset/carta-tecnica-regionale-2002-2003-5k-roma

R1: Line147 …. Finally, a mean TRW sample chronology

Change to …. Finally, a mean TRW chronology…

AR: the suggestion has been accepted and the text corrected.

R1: Line 149 Each comparison is based on …

Change to …was based on…

AR: the suggestion has been accepted and the text corrected.

R1: Figure 5 – Can you add a scale bar (in kilometres)? Or somewhere in the text mention approximate distance between the area of timber source and Rome

AR: At the end of the discussion a mention of the distance (over 1700 km) has been inserted. We guess that this observation has improved the text and we thank the Reviewer 1.

Reviewer #2 (R2): I found this an interesting, informative paper that set the results in good context, allowing the reader to understand the importance of the findings to wider fields of study. It is well written. I have a minor problem with the specificity of the conclusion that trees were felled between 40 and 53 CE - which, unless I am missing something, is not fully supported by the evidence in Table 2. If one takes the quoted sapwood range of 10-30, then C1 could go to 55CE, C17 to 58CE and C37 to 54CE, and given that sapwood estimates vary a little geographically, I think it unwise to stick rigidly to 53CE as the end point - would it not be safer to say 60CE, but then comment about any justification you may have for trimming that outer point?

AR: We approve this comment. We based our estimation on simple calculations which probably do not represent well the possible dates. We agree to change our estimation from 40 to 60 CE. The text has been changed accordingly.

R2: line 89 - the ref to Corona 1974 does not fit comfortably in the sentence as it stands, needs a little editing so that it is clear just what is meant - presumably Corona - in a rather old reference now, states how difficult results are to obtain in Rome? Suggest this is re-worded.

AR: We agree with this observation. The citation of Corona refers to an old attempt of dating a single board found in the Colosseum. This citation is useless here and has been removed.

R2: line 160 Cerris? are you talking Q cerris? Unclear.

AR: We refer to the taxa “section”, not species. The “section” Cerris, to which belongs Q. cerris, includes also Q. trojana and Q. aegilops. According to Cambini (1967), it shows anatomical features slightly different from section Robur (Q. robur, Q. petraea, Q. pubescens, Q. farnetto). We just tried to identify the section of the samples because it may be of some interest about the origin of the timber (some species are only Mediterranean).

R2: Fig 4 - presumably all DATED individual Roman oak series?

AR: the suggestion has been accepted and the text corrected.

R2: Throughout - use decimal point in t values, 6.57 not 6,57 (inconsistent at present)

AR: the suggestion has been accepted and the text corrected.

R2: The following references are used in the text but do not appear in the References list:

Janssen et al 2017 AR: inserted.

Eackstein and Bauch 1969 AR: inserted.

Momigliano and Schiavona 1988 AR: changed in “Momigliano, 2016”.

Rea 2001 AR: inserted.

Reviewer #3 (R3): Highly interesting paper - the first scientific evidence of Roman timber trade from north-eastern France to Rome and solid confirmation of Roman long-distance timber trade. This is a confirmation of high level of wood technology and wood-working skills of the Romans, and their enormous organizational level. A few of my suggestions of minor improvements concern the introductory part:

R3: - page 3, line 71: reference to Buengten et al. 2018 is not the happiest in this context. There were numerous scholars before, who stressed annual precision of dendrochronological dating - Douglass, Huber, Becker, Eckstein, etc.;

AR: we perfectly agree with R3. We know that we can start from the real beginning of dendrochronology (Douglass…). Anyway, we would like to stress the recent applications of this principle, which sometimes is still questioned. This is a very important issue: a lot of scientists of other disciplines still continue to have doubts about the effectiveness of dendrochronology. This citation demonstrates its effectiveness at a planetary level.

R3: - page 4, line 84: Guibal focused his research on shipwrecks found south of the Alps;

AR: the suggestion has been accepted and the text corrected.

R3: - page 4, lines 88-89: Corona's investigation published in 1974 was not very successful, and had nothing to do with this study.

AR: we accept this suggestion. For a clarification, see the answer to the R2. The citation has been cancelled.

R3: In general I would like to suggest to be more careful with references, and to reduce their number. Not every sentence must be supported by references.

AR: we do agree with this sentence!

We accepted the suggestion and we removed the citations redundant or unuseful.

Citations removed:

- Corona, 1974

- Haneca et al, 2009

- Hughes et al., 1981

- Jansma et al., 2014

- Sass-Klassen et al., 2008

- Schweingruber 1990

- Wigley et al., 1984.

Final considerations

We are grateful to the reviewers, which with their comments, suggestions and corrections helped us to improve our manuscript.

Best regards

 San Michele all’Adige, 29/08/2019

 Mauro Bernabei

(On behalf of all of the co-Authors)

---

## [Editor Report · Decision Letter 1]

7 Oct 2019

DENDROCHRONOLOGICAL EVIDENCE FOR LONG-DISTANCE TIMBER TRADING IN THE ROMAN EMPIRE

PONE-D-19-20970R1

Dear Dr. Bernabei,

We are pleased to inform you that your manuscript has been judged scientifically suitable for publication and will be formally accepted for publication once it complies with all outstanding technical requirements.

With kind regards,

Michal Bosela, Ph.D.

Academic Editor

PLOS ONE
---

## [Editor Report · Acceptance letter]

9 Oct 2019

PONE-D-19-20970R1 

Dendrochronological evidence for long-distance timber trading in the Roman Empire 

Dear Dr. Bernabei:

I am pleased to inform you that your manuscript has been deemed suitable for publication in PLOS ONE. Congratulations! Your manuscript is now with our production department. 

With kind regards,

on behalf of

Dr. Michal Bosela 

Academic Editor

PLOS ONE